# Loose Morphology and High Dynamism of OSER Structures Induced by the Membrane Domain of HMG-CoA Reductase

**DOI:** 10.3390/ijms22179132

**Published:** 2021-08-24

**Authors:** Ricardo Enrique Grados-Torrez, Carmen López-Iglesias, Joan Carles Ferrer, Narciso Campos

**Affiliations:** 1Centre for Research in Agricultural Genomics (CRAG) CSIC-IRTA-UAB-UB, Department of Molecular Genetics, Campus UAB, Bellaterra (Cerdanyola del Vallès), 08193 Barcelona, Spain; regrados.old@umsa.bo; 2Scientific and Technological Centers, University of Barcelona, 08028 Barcelona, Spain; c.lopeziglesias@maastrichtuniversity.nl; 3Microscopy CORE Lab, Maastricht Multimodal Molecular Imaging Institute, Maastricht University, 6229 ER Maastricht, The Netherlands; 4Department of Biochemistry and Molecular Biomedicine, Faculty of Biology, University of Barcelona, 08028 Barcelona, Spain; jcferrer@ub.edu

**Keywords:** HMG-CoA reductase, HMGR, HMGR vesicle, ER-HMGR domain, mevalonate, endoplasmic reticulum, OSER, high-pressure freezing, chemical fixation

## Abstract

The membrane domain of eukaryotic HMG-CoA reductase (HMGR) has the conserved capacity to induce endoplasmic reticulum (ER) proliferation and membrane association into Organized Smooth Endoplasmic Reticulum (OSER) structures. These formations develop in response to overexpression of particular proteins, but also occur naturally in cells of the three eukaryotic kingdoms. Here, we characterize OSER structures induced by the membrane domain of *Arabidopsis* HMGR (1S domain). Immunochemical confocal and electron microscopy studies demonstrate that the 1S:GFP chimera co-localizes with high levels of endogenous HMGR in several ER compartments, such as the ER network, the nuclear envelope, the outer and internal membranes of HMGR vesicles and the OSER structures, which we name ER-HMGR domains. After high-pressure freezing, ER-HMGR domains show typical crystalloid, whorled and lamellar ultrastructural patterns, but with wide heterogeneous luminal spaces, indicating that the native OSER is looser and more flexible than previously reported. The formation of ER-HMGR domains is reversible. OSER structures grow by incorporation of ER membranes on their periphery and progressive compaction to the inside. The ER-HMGR domains are highly dynamic in their formation versus their disassembly, their variable spherical-ovoid shape, their fluctuating borders and their rapid intracellular movement, indicating that they are not mere ER membrane aggregates, but active components of the eukaryotic cell.

## 1. Introduction

Eukaryotic HMG-CoA reductase (HMGR) has a key regulatory role in the mevalonate pathway for isoprenoid biosynthesis [1,2]. Isoprenoid products derived from this pathway are required for many diverse essential functions, including membrane biogenesis (sterols), control of growth and development (steroid hormones and plant cytokinins), protein prenylation (farnesyl and geranyl groups), protein glycosylation (dolichols) and respiration (ubiquinones) [3]. In plants, the mevalonate pathway also provides a wide variety of secondary metabolites required for defence against herbivores and pathogens or for the attraction of beneficial organisms [4]. In all plant species, HMGR is encoded by a multigene family. This was first proposed after analysis of few model plants [5], but has been confirmed by high throughput sequencing of an ever-increasing number of genomes [6]. In *Arabidopsis thaliana*, two genes (*HMG1* and *HMG2*) encode three HMGR isoforms (HMGR1S, HMGR1L and HMGR2) [5,7]. It has been suggested that different variants of plant HMGR are physically associated with other enzymes, forming metabolons for the synthesis of particular isoprenoid products, and that these metabolons would be located at particular sites of the endomembrane system [8]. The association of sterol biosynthetic enzymes at the ER membrane (one of the branches of the isoprenoid pathway) has been shown in yeast, mammals and plants [9,10,11], but no proof of the existence of metabolons involving HMGR has yet been provided.

HMGR is composed of an N-terminal membrane domain, with low or no sequence similarity among eukaryotic kingdoms, and a highly conserved catalytic domain [12,13,14]. In plant HMGR, the membrane domain has only two hydrophobic segments, whereas in yeast and animal HMGR eight membrane-spanning regions have been predicted [12,15,16]. The three *Arabidopsis* HMGR isoforms are primarily targeted to the ER by the two hydrophobic sequences of the membrane domain that interact specifically with the Signal Recognition Particle [12,17]. However, immunolocalization whole-mount studies in *Arabidopsis* cotyledon suggested that endogenous HMGR mostly localizes within spherical vesicular structures, which were therefore named HMGR vesicles [18,19]. It is not known how the integral membrane protein HMGR reaches the inside of vesicular structures, nor what relationships exist between these vesicles and the ER. 

Despite diverging evolution, the membrane domain of HMGR from the three eukaryotic kingdoms has the common capacity to induce massive proliferation of ER membranes that subsequently constitute Organized Smooth Endoplasmic Reticulum (OSER) structures [19,20,21]. When examined by transmission electron microscopy (EM), OSER structures contain tightly associated ER membranes according to three different patterns: ordered arrays with hexagonal or cubic symmetry (crystalloid ER), concentric layers (whorled ER) or simply stacked (lamellae or perinuclear karmellae) [21,22,23]. A highly conserved N-terminal motif of plant HMGR is required for OSER biogenesis [19], but no equivalent sequence has been identified in yeast or animal HMGR nor has the morphogenic mechanism been described. In *Arabidopsis*, OSER structures induced by the membrane domain of HMGR1S fused to GFP (1S:GFP chimera) also accumulate high amounts of endogenous HMGR and, therefore, have been named ER-HMGR domains [19].

Highly proliferated ER with ordered repetitive patterns was first described in the 1960s, as naturally occurring in diverse cell types from animals and plants [24,25,26,27,28,29,30] and as readily developing upon exposure to drugs [31,32]. Since then, diverse forms of hypertrophied ER have been identified in many natural and induced systems and referred to with a variety of terms, such as *cotte de mailles* [33], *paracrystalline arrays* [34], *elaborate rings of granular ER* [35], *double membrane arrays* [36], *tubuloreticular structures* [37], *undulating membranes* [38], *membrane lattice* [39], *stacks of flattened smooth ER* [40], *interlaced smooth surfaced tubules* [41], *compact areas of smooth membranes* [42], *paracrystalline ER* [43], *crystalloid membranes* [44], *organized smooth endoplasmic reticulum* [45] or *cubic membranes* [46]. An exhaustive review [47], with about 200 examples, reported that *cubic membranes* (OSER structures) are broadly distributed in the three eukaryotic kingdoms. These structures are found in numerous cell types under certain physiological conditions or appear in response to stress or disease [47]. However, in most of the aforementioned studies, images were obtained by transmission EM after chemical fixation [47]. Alternative preparation and observation techniques are necessary to further expand our knowledge on OSER ultrastructure.

In this work we first study in more depth the subcellular location of *Arabidopsis* HMGR and, particularly, the HMGR vesicles. We also characterize ER-HMGR domains in *Arabidopsis* and *Nicotiana* cells, focusing on their biogenesis, ultrastructure and dynamism. Our EM analyses uncover differences in OSER ultrastructure because of the fixation method. We find that the ER-HMGR domains are flexible live entities, fully integrated in ER architecture and dynamism.

## 2. Results

### 2.1. Subcellular Location of Arabidopsis HMGR in WT and 1S:GFP Transgenic Plants

Immunolocalization whole-mount studies in *Arabidopsis* cotyledon indicated that endogenous HMGR mostly localized inside HMGR vesicles ranging from 0.2 to 0.6 µm in diameter [18]. These studies were done with a crude rabbit polyclonal antibody raised against the catalytic domain of *Arabidopsis* HMGR1 (Ab-CD1), but it was later reported that this serum cross-reacts with *E. coli* proteins [48]. Before proceeding with a deeper localization analysis of HMGR, we wanted to confirm the whole-mount studies with an immunopurified fraction of the antibody (Ab-CD1-i) [48]. In this improved assay, we confirmed that *Arabidopsis* HMGR mostly localizes in vesicular structures of parenchymal cells, in close proximity with chloroplast (Figure 1a–c). Our observations suggest that the HMGR vesicles can measure up to 2 µm in diameter, somewhat more than previously reported.

We subsequently performed immunochemical transmission EM studies of the HMGR vesicles, both in wild type (WT) and 1S:GFP-overexpressing *Arabidopsis* plants. We used the immunopurified serum Ab-CD1-i to detect endogenous HMGR, and a commercial antibody against GFP (Ab-5450) to detect the chimeric 1S:GFP. We found selective deposition of gold particles on the surface and the inside of the HMGR vesicles, denoting the presence of both endogenous HMGR and the 1S:GFP chimera (Figure 1d–f,l). The HMGR vesicles were associated in small groups connected by ER strands (Figure 1e,f,l). This connecting ER was also immunolabeled with the antibodies against HMGR and 1S:GFP (Figure 1e,l). The ultrastructural analysis uncovered that the HMGR vesicles were delimited by an ER membrane (Figure 1e). In addition, they possessed internal ER membranes (Figure 1d,f). Both the outer and internal membranes were recognized by the Ab-CD1-i (Figure 1d,e,l) and Ab-5450 (Figure 1f,l) antibodies. These results provide a rational explanation for the whole-mount detection of HMGR protein within HMGR vesicles.

As previously reported, overexpression of 1S:GFP in transgenic *Arabidopsis* plants induces ER proliferation and OSER structure biogenesis [19]. Whole-mount and immunochemical transmission EM analyses demonstrate colocalization of the 1S:GFP chimera and high levels of endogenous HMGR in the OSER formations (Figure 1g,l). Because of the presence of HMGR protein, we name them ER-HMGR domains. They have a disordered and heterogeneous crystalloid pattern, but with a characteristic layer of large loops in their external face and more compressed structures in the inside (Figure 1h,i,l). Precise deposition of immunogold particles indicates an abundance of 1S:GFP chimera and endogenous HMGR in the ER strands of ER-HMGR domains, both in the distended external loops and the internal membrane aggregates. High levels of the 1S:GFP chimera and endogenous HMGR were also detected in the ER network (Figure 1j,l) and nuclear envelope (Figure 1j,m). Few immunogold particles were observed inside the nucleus (Figure 1j), whereas no immunolabelling was found in the Golgi apparatus (Figure 1j), mitochondria (Figure 1h) or chloroplast (Figure 1d,h,j). In the negative control, no labelling was obtained without Ab-CD1-i and Ab-5450 primary antibodies (Figure 1k).

### 2.2. Reversible Formation of ER-HMGR Domains

To further characterize the biogenesis of ER-HMGR domains, we induced the transient expression of 1S:GFP in *Nicotiana benthamiana* leaves. The agroinfiltration approach allowed generalized and abundant expression of the 1S:GFP construct in leaf epidermis (Figure 2a). At day two after transfection, massive ER proliferation led to formation of OSER structures in the transfected tissue that were detectable even at low magnification (Figure 2a). Many small OSER structures appeared at ER network junctions and a single large OSER was formed around the nuclear envelope (Figure 2c,d). The 1S:GFPm construct, containing monomeric GFP, similarly induced small OSER structures at the network junctions and a large OSER aggregate around the nucleus (Figure 2b). As previously reported [19], this indicates that the membrane domain of *Arabidopsis* HMGR, and not its dimerizing GFP partner, induces ER proliferation and membrane association into OSER. In contrast to 1S:GFP, the 1S:GFPm chimera also accumulated in hypertrophied ER strands. Thick ER strands are usually present in epidermal cells and can be visualized with the ER-GFP luminal marker (Figure 2e), but become more prominent in the case of 1S:GFPm (Figure 2b). At day six after transfection, the expression of 1S:GFP was severely reduced. Concomitant with that, OSER structures virtually disappeared, with only some remnants around the nucleus and in the cytosol (Figure 2f). The resulting ER had the usual thick strands, although broad cisternae replaced the fine network (Figure 2f). ER cisternae are common in epidermal cells transfected with the luminal ER-GFP marker, although they have a smaller size (Figure 2e). Our observations in *Nicotiana* epidermal cells indicate that OSER biogenesis is reversible. The OSER structures are not a permanent consequence of transfection with 1S:GFP or 1S:GFPm, but can be replaced by quite normal ER when the levels of the chimeric protein decrease.

### 2.3. ER-HMGR Domains Are Highly Dynamic

To further inspect OSER structure morphology and dynamism, we obtained transgenic *Arabidopsis* plants stably expressing the 1S:GFP construct. A panoramic view of seedling root epidermis showed high expression of the 1S:GFP chimera accumulating at the ER (Figure 3a). The transgenic construct induced large OSER structures (up to 10 µm in diameter) around the nuclei and smaller OSER formations at ER network junctions (Figure 3a and Appendix A). As previously reported [49,50], the ER network is highly dynamic with continuous strand movement and fusion or fission events. We found that OSER structures are connected to the ER network and participate in its dynamism. Many strands of the ER network associate with OSER formations (Figure 3b). The ER strands rapidly connect to, slide along or separate from the OSER surface (Appendix A). Small OSER structures migrate along fine or thick ER strands, whereas the large OSER formations have a more limited motion (Appendix A). In the nuclear OSER, this movement may imply a brief separation from the nuclear envelope (Appendix A). OSER structures have spherical-ovoid shapes with slight continuous variation (Appendix A). The OSER borders are not sharp, but have a fluctuating blurry aspect (Figure 3b,c), suggesting the incorporation or emergence of GFP-labelled material (likely membranes) in the OSER surface (Appendix A). We conclude that, in *Arabidopsis* cells, OSER structures are highly dynamic entities. They change in shape, have a moving surface and migrate intracellularly.

### 2.4. The Fixation and Dehydration Method Severely Affects OSER Ultrastructure

Our above confocal microscope observations of ER-HMGR domains do not fit the concept of OSER structures as rigid entities. Their tight, repetitive pattern, obtained after chemical fixation, contrasts with the flexibility and dynamism of OSER structures. To capture single states of OSER change at the ultrastructure level, we expressed 1S:GFP in *Nicotiana* leaf and submitted samples to high-pressure freezing (HPF) followed by freeze-substitution, to finally observe epidermal cells by transmission EM. We compared the HPF results with those of chemical fixation and subsequent dehydration at room temperature. With either method, OSER structures show a combination of crystalloid, lamellar and whorled membrane patterns (Figure 4a,b,g,h,j). In both cases, there is also a coincidence in cytosolic and luminal spaces. Luminal spaces correspond to the continuous internal cavity of the ER and have an electron-lucent aspect at transmission EM (Figure 4b,f,i,k). Cytosolic spaces result from the apposition of adjacent ER membranes and have a quite constant width, which is about 10–15 nm in both chemical and HPF fixation (Figure 4b,f,i,k). Cytosolic spaces have a darker aspect than luminal spaces, probably reflecting the presence of proteins that mediate the intermembrane attachment. 

In spite of the above-mentioned coincidences, OSER structures have quite different overall morphology depending on the fixation method. The most remarkable feature of OSER formations after chemical fixation is the presence of highly repetitive convoluted membranes, which have a very different smooth and turgid aspect in HPF images (Figure 4a,c,e,h,i). The convoluted pattern is exclusive to chemically fixed crystalloid domains, whereas aligned membranes are present in whorled domains, both with HPF and chemical fixation (Figure 4b,j,k). However, the crystalloid (Figure 4a,c,d) and whorled (Figure 4b) domains are far looser after HPF than after chemical fixation (Figure 4h,j). The morphological heterogeneity generated by fixation is not due to the cytosolic spaces, which are always narrow and uniform, but occurs in the luminal spaces (Figure 4b,f,i,k). The small size and regularity of luminal spaces obtained after chemical fixation contributes to the repetitive pattern (Figure 4h,i). In contrast, HPF results in larger turgid luminal spaces, which are very variable in size and morphology (Figure 4e,f). After HPF and subsequent embedding by two alternative methods, the OSER morphology is disordered and heterogeneous, but has the above-mentioned row of large loops at the periphery and is internally more compact (see similarities of Figure 4a,c,d with Figure 1h,i,l). 

To confirm the OSER ultrastructure resulting from chemical fixation, we expressed the membrane domain of *Arabidopsis* HMGR1S fused to monomeric GFP (chimera 1S:GFPm) in *Nicotiana* cells. Constructs 1S:GFP and 1S:GFPm differ in just one amino acid residue [19]. The 1S:GFPm chimera generated crystalloid, lamellar and whorled patterns with constant cytosolic distance between membranes and also regular luminal spaces (Figure 4l–o), similar to those obtained with 1S:GFP (Figure 4h–j). However, in OSER structures derived from 1S:GFPm the core of crystalloid domains was usually looser than the peripheral parts (Figure 4m). This was not observed in OSER structures derived from 1S:GFP, even when they were much larger (Figure 5e). We conclude that the dimerizing capacity of GFP may influence OSER membrane compaction during the chemical fixation process.

We also examined OSER ultrastructure in the emerging true leaves of the 1S:GFP *Arabidopsis* seedlings (10-day-old transgenic plants). We observed OSER structures with crystalloid, lamellar and whorled membrane patterns in parenchymal cells of both HPF and chemically fixed samples (Figure 5a,c,f). As mentioned above for *Nicotiana*, in *Arabidopsis* cells the three OSER ultrastructural patterns had a more relaxed morphology after HPF than after chemical fixation (Figure 5a,b,e,f). This is due to larger and more variable luminal spaces in HPF than in chemical fixation samples, whereas the dark cytosolic spaces had a similar width in both techniques (Figure 5b,d,g–i). The convoluted patterns observed in crystalloid domains after chemical fixation were absent in samples obtained by HPF (Figure 5d,g). Since HPF and chemical fixation were performed in parallel from the same samples, we conclude that the morphological differences of 1S:GFP-containing OSER were generated during the fixation process. Equivalent results were obtained in the two assayed systems, transfected *Nicotiana* leaves and *Arabidopsis* seedlings. As HPF immobilizes water and prevents its loss from the sample, the resulting transmission EM images are likely more similar to native OSER than those obtained by chemical fixation. In addition, the results obtained by HPF are more consistent with a dynamic view of the ER-HMGR domains observed with the confocal microscope.

## 3. Discussion

We found that *Arabidopsis* HMGR has several subcellular locations, such as the ER network, the nuclear envelope, HMGR vesicles, and the hypertrophied ER-HMGR domains. However, all these compartments are morphological variations of the ER. Therefore, our immunochemical results uncover that the primary targeting site of *Arabidopsis* HMGR is also its final subcellular destination. This protein does not migrate through the endomembrane system. Our observations underline the importance of the membrane domain of HMGR, which not only determines its primary and final destination sites, but also induces ER proliferation and OSER biogenesis. The resulting OSER structures (ER-HMGR domains) accumulate high levels of the chimeric 1S:GFP and endogenous HMGR ([19] and this work). Our transient expression assays indicate that the ER-HMGR domains disappear when the expression of 1S:GFP decreases, thus re-establishing normal ER morphology. A similar observation was made in transgenic *Arabidopsis* plants in which 1S:GFP expression suffered silencing after a few weeks of development [19]. Hence, in non-transfected cells natural OSER could also form or disappear, depending on the expression level of endogenous OSER-inducing proteins.

The HMGR vesicles have an outer membrane enclosing membrane material and the whole structure remains connected to the ER network. Both the internal and surrounding membranes are recognized by antibodies against HMGR and 1S:GFP, which denote ER identity. We therefore conclude that the HMGR vesicles are membrane aggregates that derive directly from the ER. The HMGR localized in the outer and the internal membranes might be metabolically active. Thus, HMGR vesicles could be a differentiated organelle of the ER involved in the synthesis of particular isoprenoid products. Additional experiments, such as metabolic labelling assays or the localization of other enzymes of the isoprenoid pathway, are required to determine whether the HMGR vesicles contain functional metabolon-like assemblies.

The HMGR domains are not a mere consequence of the accumulation of proliferated membranes in the cytosol, but behave as live structures. The OSER aggregates are an integral part of the ER network and participate in its dynamism. Small ER-HMGR domains migrate along ER strands, whereas the large ones have a more limited motion. The large OSER formations reversibly establish dynamic connections to the ER network. The ER-HMGR domains show a slow but steady change in shape, alternating between ovoid and spherical forms. Particularly intriguing are the fluctuating borders of ER-domains, which suggest continuous incorporation or release of 1S:GFP-containing material. This interpretation is reinforced by ultrastructural and immunolabelling EM studies, showing that the ER-HMGR domains have a layer of large membrane loops with 1S:GFP that surround the more compressed core. Such a morphology suggests that the OSER periphery is a site for membrane compaction or unfolding. Since samples were prepared from growing *Arabidopsis* and *Nicotiana* leaves engaged in high expression of 1S:GFP and ongoing OSER biogenesis, the moving and static images likely indicate incorporation of new membranes to the OSER aggregate. Therefore, our results suggest that crystalloid OSER structures incorporate ER membranes in their external face with subsequent compression (reduction of luminal spaces) to the inside.

Our finding that OSER structures are loose, dynamic and flexible entities could explain previous observations. The OSER-inducing chimera cytochrome b(5)-GFP was highly mobile within OSER structures and diffused rapidly between these formations and the ER network [45]. The stacked membrane associations (karmellae) produced by HMGR overexpression in yeast did not interfere with protein transit from the ER to the Golgi apparatus [51]. Overexpression of rat liver aldehyde dehydrogenase in monkey COS-1 cells induced OSER formations, but the resulting crystalloid ER did not disturb protein transport from the ER to plasma membrane or lysosomes [44]. In hamster UT-1 cells, the G protein from vesicular stomatitis virus entered and egressed from OSER aggregates freely, without apparent restriction [52]. This fast transit of proteins across OSER formations seems difficult to reconcile with the small luminal spaces and apparently rigid architecture deduced at EM after chemical fixation. The above observations are more consistent with the broad luminal spaces of OSER structures obtained by HPF and the high flexibility and dynamism seen by live imaging with the confocal microscope.

We have analysed OSER ultrastructure by transmission EM after either chemical fixation or HPF. In both cases, OSER formations are similarly composed of crystalloid, lamellar and whorled patterns, indicating that this structural diversity is not generated during fixation, but is present in the original aggregates. The two fixation methods are also reproducible in the narrow cytosolic spaces between OSER membranes. The cytosolic spaces are dense for electrons in both methods and have a constant width. These two features may suggest that membrane apposition at a fixed distance is mediated by specific proteins. In morphological free-fracture EM analysis of animal crystalloid OSER structures, a high density of homogenous particles were protruding from the cytosolic side of the internal membranes [53]. These particles may correspond to intermembrane protein bridges, the components of which should be highly abundant in the crystalloid domain.

Cryoimmobilization is regarded as the most reliable fixation strategy for ultrastructural analysis by transmission EM [54]. Diverse artefacts, particularly affecting cellular membranes, are produced by chemical fixation. It was believed that gram-positive bacteria possessed mesosomes in their plasma membrane, but it was later found that such folded invaginations are not natural but produced by chemical fixation [55,56,57]. Similarly, in sea-urchin eggs, glutardehyde fixation caused the formation of large membrane vesicles at the initial site of fusion between plasma and granule membranes [58]. More recently, in a human cell-line transformed with Epstein-Barr virus, aldehyde fixation notably reduced endosomal volume, without affecting the length of its outer membrane [59]. Thus, chemically fixed early endosomes became irregular ovoid bodies with broad protruding tubules, whereas after cryofixation the same organelles were round and turgid with an incidental short tubule [59]. It was proposed that this endosome shrinkage was produced by dehydration during chemical fixation [59]. Chemical fixation could similarly dehydrate the luminal spaces of OSER structures, without affecting membrane length. This would produce an overall shrinkage in the OSER formation, concomitant with membrane accommodation in convoluted shapes. We reproducibly obtained the convoluted membranes after chemical fixation, but never after HPF, although both methods were applied to equivalent *Arabidopsis* or *Nicotiana* samples. It is somehow amazing that a crystal-like pattern can develop from disordered and turgid membranes by the mere application of chemical fixative agents. However, it would be even more surprising that OSER morphologies having smooth turgid membranes, heterogeneous luminal spaces and large peripheral loops were generated in HPF treatment from a hypothetical native convoluted pattern. We therefore propose that the OSER ultrastructure observed after HPF is closest to the original morphology existing in vivo.

OSER structure depictions have been obtained by transmission EM from a large variety of chemically fixed eukaryotic cells and tissues, always with similar convoluted shapes in crystalloid domains [47]. Therefore, we believe that a common feature of OSER membranes may guide their compaction during chemical fixation. The aggregated lipid bilayers are highly enriched in the OSER-inducing protein and few other uncharacterized polypeptides [60,61]. For instance, not only 1S:GFP but also endogenous HMGR accumulate at high levels in ER-HMGR domains ([19] and this work). Any of these proteins could self-associate, GFP as a dimer or HMGR as a tetramer [62,63]. Highly abundant self-associating proteins could then provide a structural frame to build the crystal-like convoluted pattern that would be established during slow dehydration and chemical cross-linking.

Crystalloid ultrastructural patterns generated by 1S:GFP were homogeneous and compact, whereas those generated by 1S:GFPm had a looser core (Figure 4g,h,m), indicating that the presence of monomeric GFP instead of the original dimerizing GFP was responsible for the difference. Since chemical fixatives rely on diffusion to penetrate the sample and are consumed during fixation, their effects are discontinuous with slower cross-linking in the inside [54]. We speculate that this temporal discontinuity, together with a lower association capacity of 1S:GFPm with respect to 1S:GFP, could reduce membrane compaction at the core of the crystalloid domain. Independently of the molecular mechanism, the heterogeneous membrane compaction with 1S:GFPm reinforces the notion that chemical fixation may illegitimately affect OSER ultrastructure.

## 4. Materials and Methods

### 4.1. Plant Material

Experiments were performed with *Arabidopsis* (*Arabidopsis thaliana*) WT Col 0 or a transgenic line overexpressing 1S:GFP in the same genetic background. The preparation of the 1S:GFP transgenic line was described previously [19]. In this line, 1S:GFP is under control of the cauliflower mosaic virus 35S promoter, which confers high expression in the whole plant. Seeds were surface sterilized by washing three times with 70% (*v*/*v*) ethanol and three times with 100% (*v*/*v*) ethanol, sowed in petri dishes with half concentrated Murashige and Skoog (MS) medium [64] and vernalized at 4 °C for 3 d. Growth was at 22–24 °C and long day conditions (16 h light/8 h darkness). 

### 4.2. Constructs

The identity and source of protein constructs was as follows: 1S:GFP and 1SGFPm, N-terminal fragment 1-178 from *Arabidopsis* HMGR1S (accession AAF16652) fused, respectively, to GFP or monomeric GFP [19]. ER:GFP, GFP sequence with appended ER signal peptide at the N-terminus and ER retention signal at the C-terminus, encoded by plasmid pVKH-GFP-HDEL [65].

### 4.3. Transient Expression in Nicotiana Benthamiana Leaves

Transient expression of 1S:GFP and 1S:GFPm in *N. benthamiana* leaves was achieved by agroinfiltration as previously described [66]. 1S:GFP and 1S:GFPm was previously cloned into plasmid pPCV002 under control of the cauliflower mosaic virus 35S promoter, which confers high constitutive expression [19]. The pPCV002 derivatives were transformed into *Agrobacterium tumefaciens* strain GV3101 pMP90RK [67]. The transformed bacteria were grown in YEB medium (per litre: 5 g beef extract, 1 g yeast extract, 5 g bacteriological peptone, 5 g sucrose, 2 mmol MgSO_4_) containing 100 µg/mL each of kanamycin, rifampicin and carbenicillin at an OD_600_ of 0.5 to 1.0. Cells were harvested by centrifugation and suspended in 10 mM *N*-morpholino ethanesulfonic acid (MES, pH 5.7), 10 mM MgCl_2_ and 0.2 mM acetosyringone to an OD_600_ of 1.0. Bacteria were infiltrated into leaves with a 1-mL disposable syringe without a needle. Expression was examined daily, until day 6 after agroinfiltration. 

### 4.4. Source and Use of Antibodies

The catalytic domain of *Arabidopsis* HMGR1S (CD1) produced in *Escherichia coli* was used as immunogen to produce a polyclonal antibody in rabbit and the resulting serum was immunosubstracted to remove IgG reacting against the bacterial proteins [48]. The immunopurified serum (Ab-CD1-i) was used as primary antibody at 1:500 for whole mount and 1:1000 for transmission EM. Anti-rabbit IgG secondary antibodies for HMGR detection were code Ab150066 (Abcam, Toronto, ON, Canada) coupled to Alexa Fluor-555 at 1:1000 for whole mount (Figure 1a,b), code Ab150068 (Abcam, Toronto, ON, Canada) coupled to Alexa Fluor-594 at 1:1000 for whole mount (Figure 1g), code 111-215-144 (Jackson Immunoresearch, Cambridge, UK) coupled to an 18-nm gold particle at 1:30 for transmission EM (Figure 1d) and code 111-205-144 (Jackson Immunoresearch, Cambridge, UK) coupled to a 12-nm gold particle at 1:30 for transmission EM (Figure 1e,f,l,m).

GFP was detected with Ab-5450 (Abcam, Toronto, ON, Canada) as the primary antibody at 1:1000 for whole mount and transmission EM. Anti-goat IgG secondary antibodies for GFP detection were code Ab150133 (Abcam, Toronto, ON, Canada) coupled to Alexa Fluor-488 at 1:1000 for whole mount, and code 705-215-147 (Jackson Immunoresearch, Cambridge, UK) coupled with an 18-nm gold particle at 1:15 for transmission EM.

### 4.5. Immunolocalization in Whole Mount

Whole-mount in situ immunolocalization was done as described [68] with modifications. After fixation in 4% (*w*/*v*) paraformaldehyde, seedlings were incubated in methanol to remove chlorophylls. Five cycles of seedling freezing and thawing on glass slides were performed to permeate tissue, followed by incubation with 2% (*w*/*v*) Driselase (D8037; Sigma Aldrich-Merk, Madrid, Spain) to allow for the penetration of antibodies through the plant cell wall. After blocking with 3% (*w*/*v*) bovine serum albumin (BSA) solution, simultaneous incubation with one or two primary antibodies (Ab-CD1-i at 1:500, Ab-5450 at 1:1000) was performed directly on the slides. Samples were then washed with phosphate-buffered saline (PBS) and incubated with the corresponding secondary antibody at 1:1000. The secondary antibody to detect HMGR was Abcam Ab150066 (Alexa Fluor-555) or Ab150068 (Alexa Fluor-594). The secondary antibody to detect GFP was Abcam Ab150133 (Alexa Fluor-488). After washing with PBS, samples were sealed for observation at the confocal microscope.

### 4.6. Confocal Microscopy

Confocal laser microscopy was performed with spectral microscope Olympus FV1000 (objectives UPLSAPO 60x O, numerical aperture: 1.35 and UPLSAPO 60x W, numerical aperture: 1.20) or Leica TCS SP5 (objectives HCX PL APO CS 40x/1.25 Oil UV, HCX PL APO 63x/1.20 W corrected for UV, and HCX PL APO CS 63x/1.20 water UV) at room temperature. Fluorophores were detected with the following excitation and emission wavelengths: GFP (excitation = 488 nm, emission = 500–545 nm), Alexa Fluor-555 (excitation = 559 nm, emission = 570–610 nm) and Alexa Fluor-594 (excitation = 559 nm, emission = 575–620 nm) and chlorophyll (excitation = 559 nm, emission = 640–680 nm). Images were acquired with the software FV10-ASW 4.1 (Olympus, Hamburg, Germany) and LAS AF 2.7.3.9723 (Leica Microsystems, Wetzlar, Germany) and processed with the software ImageJ 1.50i (http://imagej.nih.gov/ij, accessed on 28 May 2020).

### 4.7. Chemical Fixation for Ultrastructural Studies

Explants from *Nicotiana* leaves or *Arabidopsis* seedlings were excised under a stereomicroscope and transferred to glass vials filled with 1.5% (*v*/*v*) paraformaldehyde and 1.5% (*v*/*v*) glutaraldehyde in 0.1 M cacodylate buffer (pH 7.4) containing 2 mM CaCl_2_. The vials were degassed briefly to allow penetration of the fixative into tissue and incubated at 4 °C for 24 h. After washing with the cacodylate-CaCl_2_ buffer without fixative, samples were post-fixed for 3 h at 4 °C with 1% (*w*/*v*) osmium tetroxide and 0.8% (*w*/*v*) K_3_Fe(CN)_6_ in the same buffer. Samples were subsequently dehydrated in acetone, infiltrated with Spurr resin for 2 d, embedded in the same resin and polymerized at 60 °C for 48 h. 

### 4.8. High-Presure Freezing (HPF) for Ultrastructural Studies

Explants from *Nicotiana* leaves or *Arabidopsis* seedlings were excised under a stereomicroscope and transferred to aluminium planchette with a 200 μm-deep cavity that was subsequently filled with yeast (*Saccharomyces cerevisiae*) paste and cryoimmobilized immediately with a high pressure freezer (EM Pact (Leica)). Freeze substitution of frozen samples was performed in an automatic freeze substitution system (EM AFS (Leica)) with acetone containing 2% (*w*/*v*) osmium tetroxide and 0.1% (*w*/*v*) uranyl acetate, for 3 d at −90 °C. On the fourth day, the temperature was raised by 5 °C per hour to room temperature. At this temperature, samples were rinsed in acetone, infiltrated with Epon (*Nicotiana* samples, Figure 4) or Spurr (*Arabidopsis* samples, Figure 5) resin for 2 d, embedded in a thin layer of the same resin and polymerized at 60 °C for 48 h.

### 4.9. Ultrastructural Analysis 

Embedded blocks for ultrastructural analysis were submitted to thin sectioning and cell integrity was confirmed at the light microscope. Ultrathin sections were obtained using an Ultracut UC6 Ultramicrotome (Leica) and mounted on Formvar-coated copper grids. Samples were stained with 2% (*w*/*v*) uranyl acetate in water and lead citrate. Samples were observed in a JEM-1010 Electron Microscope (JEOL) equipped with a CCD Camera SIS Megaview III and the AnalySIS software. After capture, image brightness and contrast were adjusted with ImageJ 1.50i for a better visualization.

### 4.10. Immunochemical Ultrastructural Analysis

Explants from *Arabidopsis* seedlings were cryoimmobilized by high-pressure freezing using an EM Pact (Leica) with yeast paste as the filler. Freeze substitution of frozen samples was performed in an automatic freeze substitution system (EM AFS (Leica)) with methanol containing 0.5% (*w*/*v*) uranyl acetate at −90 °C for 3 d. On the fourth day, the temperature was raised by 5 °C per hour to −50 °C. At this temperature, samples were rinsed in acetone, infiltrated and flat embedded in Lowicryl HM20 for 4 d. Ultrathin sections were picked up on Formvar-coated nickel grids. Sample-containing grids were incubated on drops of PBS with 5% (*w*/*v*) BSA for 20 min at room temperature. After removal of the washing solution, drops of PBS with the primary antibody (Ab-CD1-i or Ab-5450 at 1:1000) and 1% (*w*/*v*) BSA were added and incubated for 2 h. Grids were washed three times for 30 min with a drop of PBS with 0.25% (*v*/*v*) Tween 20 and incubated for 1 h in drops of PBS with the secondary antibody and 1% (*w*/*v*) BSA. Secondary antibodies (at 1:30) for HMGR detection were codes 111-205-144 (12-nm gold particle) or 111-215-144 (18-nm particle) from Jackson Immunoresearch (Cambridge, UK). The secondary antibody for GFP detection (code 705-215-147; 18-nm particle; Jackson Immunoresearch, Cambridge, UK) was used at 1:15. The grids were washed three times with a drop of PBS for 5 min and two times with distilled water and air-dried. In control assays for the nonspecific binding of the gold-conjugated antibody, the primary antibody was omitted. Sections were stained with 2% (*w*/*v*) uranyl acetate in water and lead citrate and observed in a JEM-1010 electron microscope (JEOL) with an SIS Mega View III CCD camera.

## Figures and Tables

**Figure 1 ijms-22-09132-f001:**
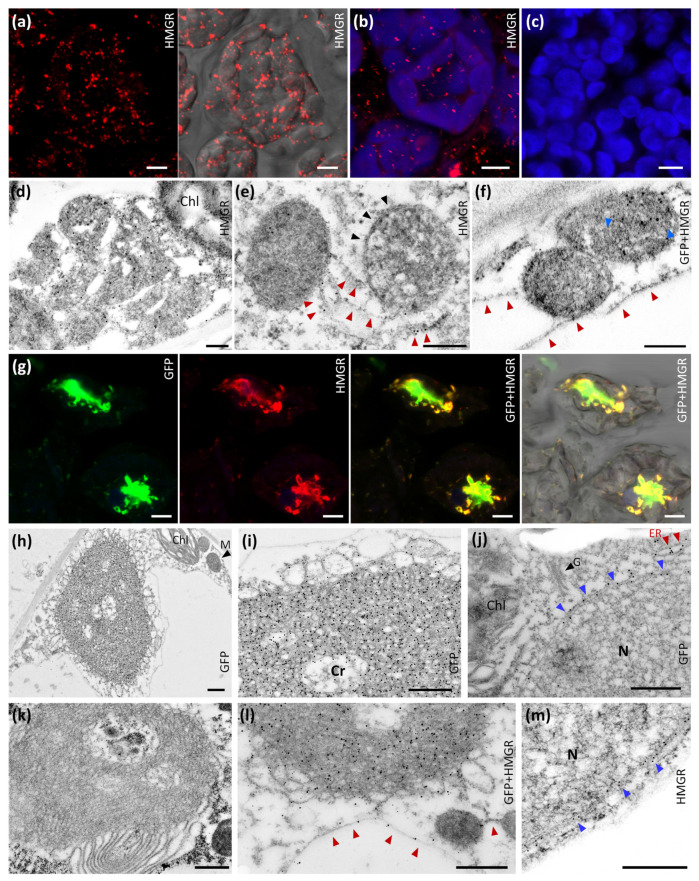
*Arabidopsis* HMGR localizes in the ER network, nuclear envelope, HMGR vesicles and ER-HMGR domains. (**a**–**c**) Whole-mount immunohistochemical analysis of cotyledon parenchymal cells from 6-day-old *Arabidopsis* WT seedlings (**a**) Immunodetection of HMGR with Ab-CD1-i and anti-rabbit IgG secondary antibody (Alexa Fluor 555, in red), visualized by confocal microscopy under dark (left) or bright fields (right). (**b**) Immunodetection of HMGR with Ab-CD1-i and anti-rabbit IgG secondary antibody (AlexaFluor 555, in red), and simultaneous detection of chlorophyll (in blue). (**c**) Negative control without the Ab-CD1-i antibody. The irregular corpuscles, 0.2 to 2 µm in length, correspond to HMGR vesicles. The elliptic bodies, 6 to 8 µm in diameter, correspond to chloroplasts. Images were obtained by Z-projection encompassing 3 (**a**), 10 (**b**) or 4 (**c**) µm in the Z-axis. Bars, 5 µm. (**d**–**f**) Immunochemical study of HMGR vesicles by transmission EM. Leaf samples from 10-day-old *Arabidopsis* WT or 1S:GFP seedlings were processed by HPF and embedded with Lowicryl HM20. (**d**) HMGR was detected in cotyledon from WT seedlings with Ab-CD1-i and anti-rabbit-IgG (18 nm particle). (**e**) HMGR was detected in true leaf from 1S:GFP seedlings with Ab-CD1-i and anti-rabbit-IgG (12 nm particle). (**f**) Double immunolocalization of HMGR and 1S:GFP in true leaf from 1S:GFP seedlings. HMGR was detected with Ab-CD1-i and anti-rabbit-IgG (12 nm particle) and 1S:GFP was detected with Ab-5450 and anti-goat-IgG (18 nm particle). Black and blue arrowheads indicate, respectively, the external and internal membranes from HMGR vesicles. Red arrowheads indicate ER strands. Bars, 250 nm. (**g**) Whole-mount immunohistochemical analysis of cotyledon parenchymal cells from 6-day-old *Arabidopsis* 1S:GFP transgenic seedlings. 1S:GFP was detected with Ab-5450 and secondary antibody Alexa fluor 488 (green). HMGR was detected with Ab-CD1-i and secondary antibody Alexa fluor 594 (red). Images were obtained by Z-projection encompassing 12 µm in the Z-axis. Bar, 5 µm. (**h**–**m**) Immunolocalization of HMGR and 1S:GFP by transmission EM. True leaves from 10-day-old *Arabidopsis* WT seedlings were processed by HPF and embedded with Lowicryl HM20. (**h**–**j**) 1S:GFP was detected with Ab-5450 and anti-goat-IgG (18 nm particle). (**k**) Negative control without Ab-CD1-i and Ab-5450. (**l**) Double immunolocalization of 1S:GFP (18 nm particle) and HMGR (12 nm particle). (**m**) HMGR was detected with Ab-CD1-i and anti-rabbit-IgG (12 nm particle). Chloroplast (Chl). ER strands (red arrowheads). Golgi apparatus (G). Mitochondria (M). Nuclear envelope (blue arrowheads). Nucleus (N). Bars, 500 nm.

**Figure 2 ijms-22-09132-f002:**
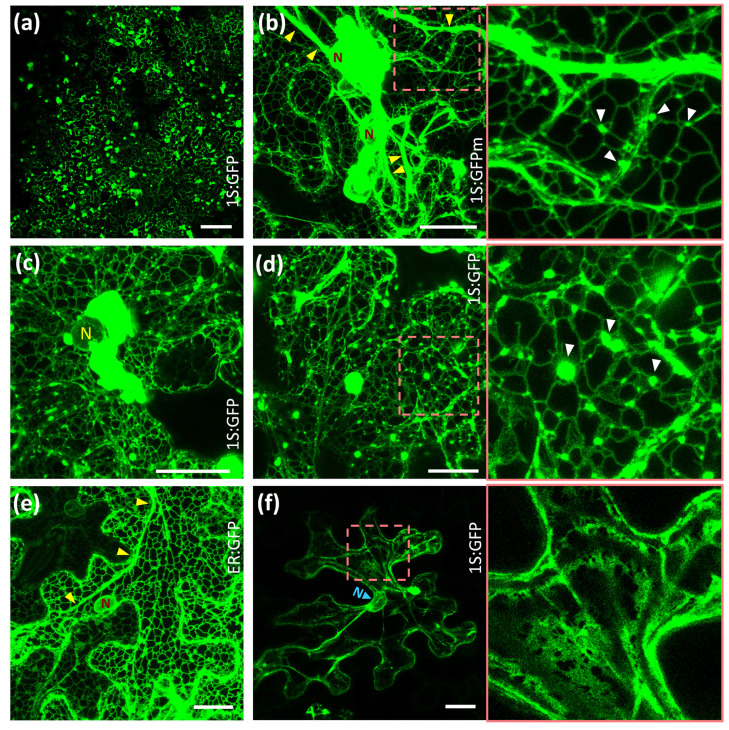
Reversible biogenesis of ER-HMGR domains in transfected *Nicotiana* cells. *Nicotiana* leaves were transfected with constructs encoding 1S:GFP (**a**,**c**,**d**,**f**), 1S:GFPm (**b**) or the control ER lumen marker ER:GFP (**e**) and epidermal cells were visualized by confocal laser microscopy. The expression time was 2 days (**a**), 3 days (**b**–**e**) or 6 days (**f**). The square regions indicated in (**b**,**d**,**f**) are shown enlarged on the right. Images are a single section (**a**) or Z-projections encompassing 10 (**b**), 21 (**c**), 7 (**d**), 12 (**e**) or 14 (**f**) µm in the Z-axis. Nucleus (N). OSER structures (white arrowheads). Thick ER strands (yellow arrowheads). Bars, 100 µm (**a**), 20 µm (**b**–**f**).

**Figure 3 ijms-22-09132-f003:**
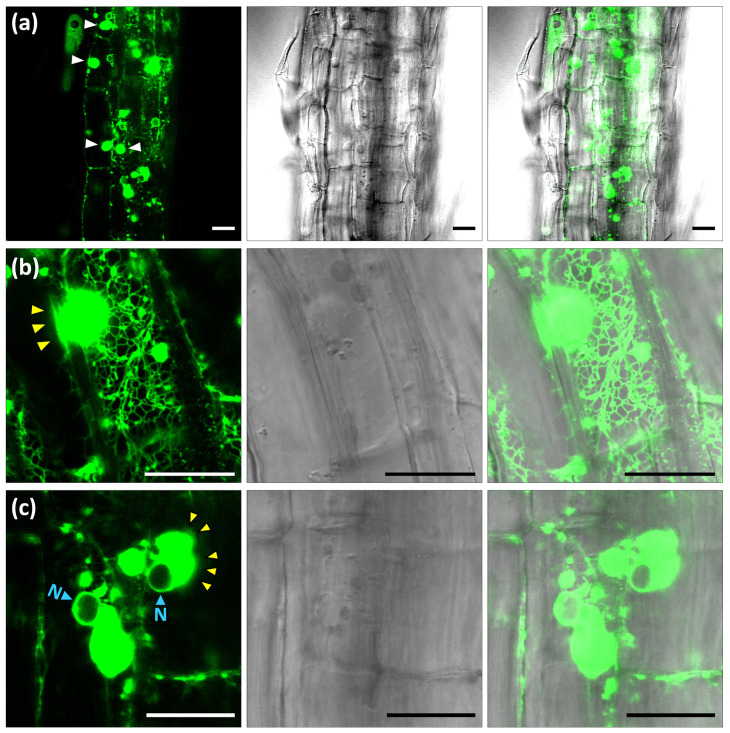
Characterization of ER-HMGR domains in *Arabidopsis* 1S:GFP transgenic plants. *Arabi**dopsis* 1S:GFP 9-day-old seedlings were analysed by confocal laser microscopy. The pictures show three channels (GFP, bright field and merge) of single sections from root epidermal cells that were subsequently characterized by live imaging: (**a**) panoramic view of root epidermis with large ER-HMGR domains indicated by white arrowheads (Appendix A); (**b**) large ER-HMGR domain with dynamic connections to the ER network (Appendix A); (**c**) nuclear ER-HMGR domains with changing spherical-ovoid shape (Appendix A). Blurry fluctuating borders (yellow arrowheads). Nucleus (N). OSER structures (white arrowheads). Bars, 20 µm.

**Figure 4 ijms-22-09132-f004:**
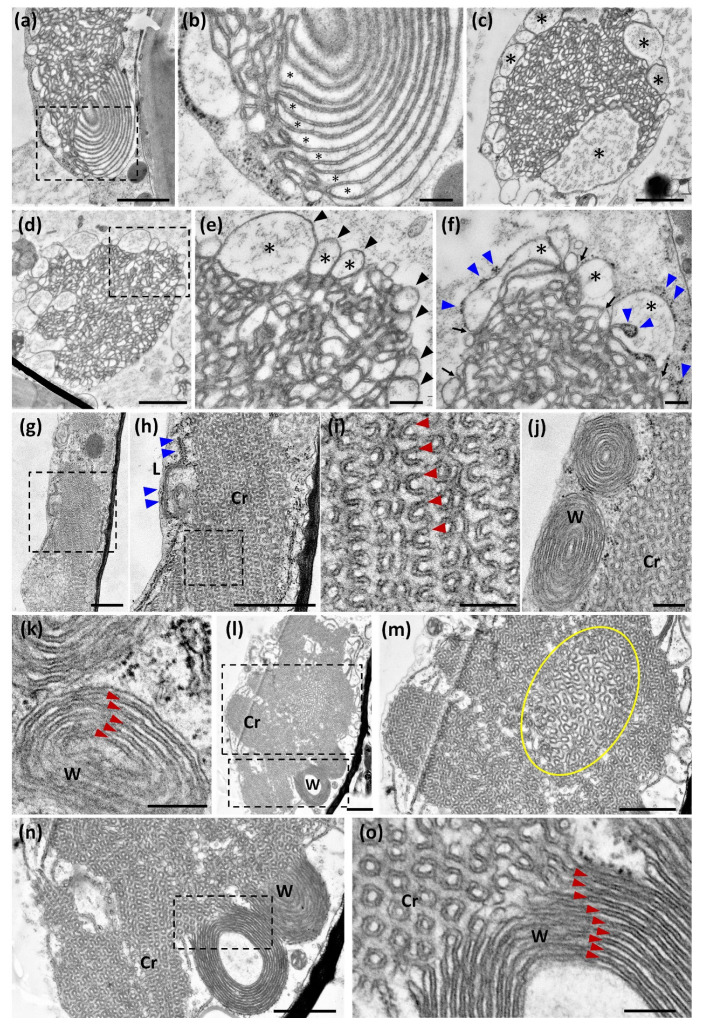
Ultrastructural analysis of ER-HMGR domains in *Nicotiana* epidermal cells. Three days after agroinfiltration with 1S:GFP (**a**–**k**) or 1S:GFPm (**l**–**o**), *Nicotiana* leaves were processed for transmission EM. (**a**–**f**) 1S:GFP-transfected leaves were submitted to HPF and embedded in Epon resin. (**a**) Panoramic view ER-HMGR domain with a crystalloid region in sagittal section and a whorled region in transversal section. (**b**) Magnified region of panel (**a**), to show the clear luminal spaces alternating with darker (denser for electrons) cytosolic spaces. (**c**,**d**) Panoramic view of crystalloid ER-HMGR domains with large loops surrounding the internal, more compact structure. (**e**) Magnified region of (**d**) with large loops. (**f**) Crystalloid structure with interspersed clear-luminal and dark-cytosolic spaces. The OSER cytosolic spaces are continuous with the cytoplasm (black arrows). Ribosomes are visible at the external face of the OSER structure. Occasionally, they are trapped in internal cytosolic spaces, near the periphery of the ER aggregate. (**g**–**k**) 1S:GFP-transfected leaves were submitted to chemical fixation and embedded in Spurr resin. (**g**) Panoramic view of crystalloid ER-HMGR domain. (**h**) Magnified region of (**g**) to show the repetitive smooth ER and a thin lamellar region on the left side. Ribosomes are excluded from the ER-HMGR domain. (**i**) Magnified regions of (**h**). (**j**) Whorled and crystalloid regions. (**k**) Whorled region with regular cytosolic and luminal spaces. (**l–o**) 1S:GFPm-transfected leaves were submitted to chemical fixation and embedded in Spurr resin. (**l**) Panoramic view of ER-HMGR domain with crystalloid, lamellar and whorled patterns. (**m**) Magnified region of (**l**) to show the looser region (circled) in the core of the ER-domain. (**n**) Magnified region of (**l**) to show the transition between whorled and crystalloid regions. (**o**) Magnified region of (**n**). Large membrane loops (black arrowheads). Crystalloid region (Cr). Cytosolic spaces (red arrowheads). Lamellar region (L). Luminal spaces (asterisks). Ribosomes (blue arrowheads). Whorled region (W). Bars, 1 µm (**a**,**c**,**d**,**g**,**h**,**l**–**n**), 200 nm (**b**,**e**,**f**,**i**–**k**,**o**).

**Figure 5 ijms-22-09132-f005:**
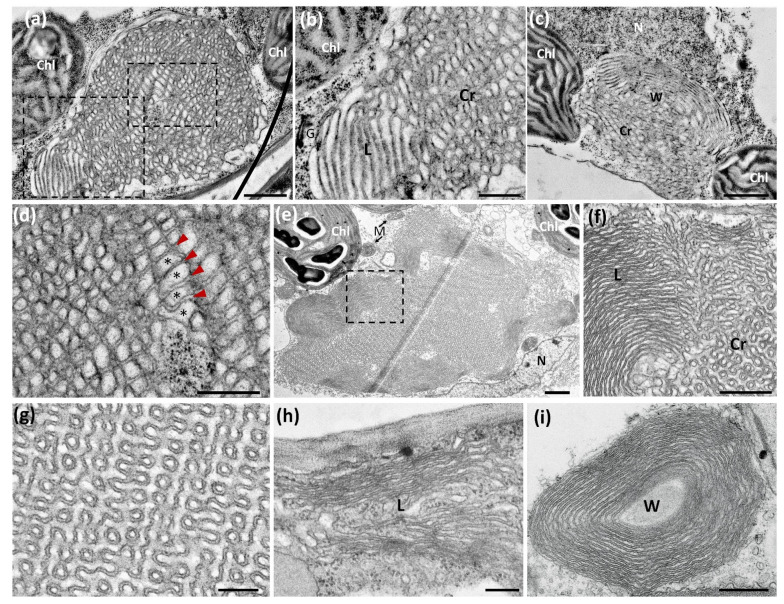
Ultrastructural analysis of ER-HMGR domains in *Arabidopsis* parenchymal cells. Emerging true leaves from *Arabidopsis* 10-day-old seedlings transgenic for 1S:GFP were submitted to HPF or chemical fixation and embedded in Spurr resin. (**a**–**d**) Samples obtained by HPF. (**a**) Panoramic view of a crystalloid-lamellar ER-HMGR domain located between chloroplasts. (**b**) Magnified region of panel (**a**), to show the transition between crystalloid and lamellar regions. (**c**) Nuclear ER-HMGR domain with crystalloid and whorled patterns. (**d**) Magnified region of (**a**)**,** to show alternation of cytosolic (dark) and luminal (clear) spaces. Notice the narrow homogenous width of the cytosolic spaces and the broader and more variable size of the luminal spaces. (**e**–**i**) Samples obtained by chemical fixation. (**e**) Panoramic view of ER-HMGR domain with crystalloid, lamellar and whorled patterns. Notice the large size of the OSER structure (12 μm long). (**f**) Magnified region of panel (**e**), to show the transition between lamellar and crystalloid regions. (**g**) Detail of a crystalloid structure to show the repetitive pattern of dark cytosolic spaces and lighter luminal spaces. (**h**) Lamellar ER-HMGR domain. (**i**) Whorled ER-HMGR domain. Chloroplast (Chl). Crystalloid region (Cr). Cytosolic spaces (red arrowheads). Golgi apparatus (G). Lamellar region (L). Luminal spaces (asterisks). Mitochondria (M). Nucleus (N). Whorled region (W). Bars, 1 µm (**a**,**c**,**e**), 500 nm (**b**,**d**,**f**,**i**), 200 nm (**g**,**h**).

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
