# Peer review of "Loose Morphology and High Dynamism of OSER Structures Induced by the Membrane Domain of HMG-CoA Reductase"

_ijms, 2021, doi:10.3390/ijms22179132_

Round 1

Reviewer 1 Report

In figure 1, not all the pictures were labelled with scale bar, Please add information to the those pictures. Similar concerns exist in figure 3.

Author Response

As requested, scale bars are now in all pictures of Figures 1 and 3.

Reviewer 2 Report

The manuscript is meant to demonstrate that OSER (Organized Smooth Endoplasmic Reticulum) structures are active components at least in plant cells. HMG-CoA reductase (HMGR) is an ER-localized key enzyme of the isoprenoid mevalonic pathway capable to induce massive ER membrane proliferation when its membrane domain is over-expressed. Thus, this protein appears as a perfect tool to investigate this problematic.

Using purified antibodies raised against the catalytical Arabidopsis HMGR1, this new study demonstrated that induced ER-HMGR domains contain endogenous HMGR in addition of the chimeric protein. The identity of the promoter used to overexpress genes should be specified in the experimental section.

Transient expression allowed to show that OSER formation is reversible however, it is not clear for the reader, if OSER cannot be observed because 1S:GFP (the tool used to observe) expression decreased after a while or OSER structures are reorganized. To restrain infiltration artefacts, it would be interesting (in the future) to create stable transformants expressing a construct under the control of an inducible promoter.

Stable transformation in Arabidopsis highlighted dynamic structures (confocal observations). Then, using TEM, they showed that dimerizing capacity of GFP may influence the compaction of OSER membranes during the chemical fixation process and concluded that HPF is more suitable for OSER observations than chemical fixation. This later point might be very useful to consider to people studying OSER structure formation in other contexts.

Author Response

[Answer to the second paragraph] The identity of the promoter driving expression of 1S:GFP in transgenic plants has been included now in text lines 508-510 (section ‘Plant Material’ of Materials and Methods). The identity of the promoter for transient expression assays appears in text lines 522-524.

[Answer to the third paragraph] The analysis of transgenic plants expressing 1S:GFP was reported previously (Ferrero et al, 2015, Plant Physiol 168, 899–914). After few weeks of plant development, 1S:GFP suffered silencing, with led to disappearance of OSER structures and re-establishment of normal ER morphology. These observations reinforce the conclusions of our transient expression assays. The results with transgenic plants have been conveniently cited and commented in text lines 396-402 (Discussion section). We thank the reviewer for the proposal.